# Identification and Validation of an *Aspergillus nidulans* Secondary Metabolite Derivative as an Inhibitor of the Musashi-RNA Interaction

**DOI:** 10.3390/cancers12082221

**Published:** 2020-08-08

**Authors:** Lan Lan, Jiajun Liu, Minli Xing, Amber R. Smith, Jinan Wang, Xiaoqing Wu, Carl Appelman, Ke Li, Anuradha Roy, Ragul Gowthaman, John Karanicolas, Amber D. Somoza, Clay C. C. Wang, Yinglong Miao, Roberto De Guzman, Berl R. Oakley, Kristi L. Neufeld, Liang Xu

**Affiliations:** 1Departments of Molecular Biosciences, the University of Kansas, Lawrence, KS 66045, USA; lan@ku.edu (L.L.); gajun988@gmail.com (J.L.); arsmith1@stanford.edu (A.R.S.); wuxq@ku.edu (X.W.); carltonapps@gmail.com (C.A.); lk_jzs@xiyi.edu.cn (K.L.); rdguzman@ku.edu (R.D.G.); boakley@ku.edu (B.R.O.); klneuf@ku.edu (K.L.N.); 2Bio-NMR Core Facility, the University of Kansas, Lawrence, KS 66045, USA; mlxing@umich.edu; 3Center for Computational Biology, the University of Kansas, Lawrence, KS 66045, USA; jawang@ku.edu (J.W.); ragul@umd.edu (R.G.); miao@ku.edu (Y.M.); 4High Throughput Screening Laboratory, the University of Kansas, Lawrence, KS 66045, USA; anuroy@ku.edu; 5Program in Molecular Therapeutics, Fox Chase Cancer Center, Philadelphia, PA 19111, USA; John.Karanicolas@fccc.edu; 6Department of Chemistry, University of Southern California, Los Angeles, CA 90007, USA; amber.somoza@gmail.com (A.D.S.); clayw@usc.edu (C.C.C.W.); 7Department of Pharmacology and Pharmaceutical Sciences, School of Pharmacy, University of Southern California, Los Angeles, CA 90007, USA; 8Department of Cancer Biology, the University of Kansas Cancer Center, Kansas City, KS 66160, USA; 9Department of Radiation Oncology, the University of Kansas Cancer Center, Kansas City, KS 66160, USA

**Keywords:** RNA-binding proteins, Musashi, drug discovery, Notch signaling, Wnt signaling, cancer therapy, fungi secondary metabolite derivative

## Abstract

RNA-binding protein Musashi-1 (MSI1) is a key regulator of several stem cell populations. MSI1 is involved in tumor proliferation and maintenance, and it regulates target mRNAs at the translational level. The known mRNA targets of MSI1 include *Numb*, *APC*, and *P21^WAF-1^*, key regulators of Notch/Wnt signaling and cell cycle progression, respectively. In this study, we aim to identify small molecule inhibitors of MSI1–mRNA interactions, which could block the growth of cancer cells with high levels of MSI1. Using a fluorescence polarization (FP) assay, we screened small molecules from several chemical libraries for those that disrupt the binding of MSI1 to its consensus RNA. One cluster of hit compounds is the derivatives of secondary metabolites from *Aspergillus nidulans*. One of the top hits, Aza-9, from this cluster was further validated by surface plasmon resonance and nuclear magnetic resonance spectroscopy, which demonstrated that Aza-9 binds directly to MSI1, and the binding is at the RNA binding pocket. We also show that Aza-9 binds to Musashi-2 (MSI2) as well. To test whether Aza-9 has anti-cancer potential, we used liposomes to facilitate Aza-9 cellular uptake. Aza-9-liposome inhibits proliferation, induces apoptosis and autophagy, and down-regulates Notch and Wnt signaling in colon cancer cell lines. In conclusion, we identified a series of potential lead compounds for inhibiting MSI1/2 function, while establishing a framework for identifying small molecule inhibitors of RNA binding proteins using FP-based screening methodology.

## 1. Introduction

Post-transcriptional gene regulation occurs at the levels of pre-mRNA splicing and maturation, as well as mRNA transport, editing, storage, stability, and translation. This level of gene regulation is essential for normal development, but when dysregulated, contributes to diseases such as cancer. Post-transcriptional gene regulation is mediated by RNA-binding proteins, which are emerging new targets for cancer therapy.

One such RNA-binding protein is Musashi-1 (Msi1/MSI1), which was first identified in *Drosophila* and studied extensively for its role in sensory organ progenitor (SOP) cell lineage establishment [1]. In *Drosophila msi1* loss-of-function mutants, a SOP cell divides to produce two daughter cells with low Notch signaling rather than one daughter cell with relatively high Notch signaling and one with low Notch signaling [1]. In mammalian systems, MSI1 is expressed in multiple stem cell populations, including adult stem cells of the breast, colon, hair follicles, and brain [2,3,4,5,6]. MSI1 is overexpressed in a wide variety of cancers [4,7,8,9,10,11,12,13,14,15]; although it’s specific function in tumorigenesis is largely unknown. Knocking down MSI1 reduces tumor progression in several cancer models [9,12] and overexpressing MSI1 in a rat intestinal crypt-derived cell line induces tumorigenesis in a mouse xenograft model [16]. These findings suggest that MSI1 can promote tumorigenesis and support MSI1 as a promising drug target for cancer therapy.

MSI1 appears to function by binding to the 3′UTR of target mRNAs and thereby regulating protein translation [17,18,19,20]. Targets whose translation is repressed by MSI1 include *NUMB*, *APC*, and *P21^WAF-1^* [17,20,21,22]. In this study, we aimed to identify small molecules that disrupt the binding of MSI1 with target mRNAs. Novel small molecule inhibitors may be used in the future for treating patients with high MSI1 expression, while also serving as a tool to elucidate MSI1′s role in cancer initiation and progression.

Previous efforts to identify small molecule inhibitors of protein-protein interactions have focused on proteins with well-defined binding pockets, while inhibitors of RNA-binding proteins were limited to proteins that bind to viral RNAs [23,24,25]. In an effort to identify small molecule inhibitors of MSI1, we employed a fluorescence polarization (FP) competition assay and screened nearly 2000 compounds, including those from three NCI (National Cancer Institute) libraries [26] and our in-house compounds. We identified a series of hit compounds including Aza-9, a semi-synthetic derivative of a secondary metabolite from *Aspergillus nidulans* [27] that inhibits MSI1 RNA binding activity. Surface plasmon resonance (SPR), nuclear magnetic resonance (NMR) spectroscopy, and RNA-IP assays demonstrate that Aza-9 binds to MSI1 directly.

Musashi-2 (MSI2) shares a high degree of sequence identity to MSI1 [28], functions redundantly in certain tissues, and is also overexpressed in many cancers [14,15,29,30,31,32]. We show here that Aza-9 binds to MSI2 as well. To test Aza-9 function in cells, we introduced liposomes to facilitate Aza-9 cellular entry. We show that with Aza-9-liposome treatment, colon cancer cells HCT-116 and DLD-1 undergo apoptosis/autophagy, cell cycle arrest and modest Notch/Wnt signaling down-regulation.

## 2. Results

### 2.1. FP Assay Optimization

We used an FP assay to identify compounds that disrupt the interaction of MSI1 with a target RNA, *Numb*. In our initial binding assays, we used 2 nM of RNA as specified in previous publications [33,34]. We measured the binding of different concentrations of protein G B1 domain-tagged MSI1 RNA-binding Domain1 (GB1-RBD1) to Fluorescein isothiocyanate (FITC)-labeled *Numb* RNA (*Numb^FITC^*) or control RNA (*CTL^FITC^*). As shown in Figure 1A, *Numb^FITC^* but not *CTL^FITC^* bound to GB1-RBD1 as indicated by the increased polarization value. We further tested binding by competing the RBD1-*Numb^FITC^* complex with non-labeled *Numb* or control RNA. *Numb* RNA displaced the *Numb^FITC^* from the protein–RNA complex at a much lower Ki than the control RNA (Figure 1B). This established binding between *Numb* RNA and GB1-RBD1 formed the basis for the following screening assay.

To adapt our initial binding assay to a more robust screening method, we carried out several optimization steps. First, we titrated both the GB1-RBD1 protein (1 to 250 nM) and *Numb^FITC^* RNA (1 to 200 nM). RNA concentrations less than 1 nM produced such low fluorescent signals that FP values were not stable. Figure 1C shows a dose response curve of RNA concentrations ≥ 1 nM. Optimal binding to RBD1 was achieved with 2–10 nM RNA, with an RBD1 equilibrium dissociation constant (Kd) value less than 250 nM. We chose 2 nM RNA for future assays based on the lowest Kd value of the protein (Figure 1C). Next, we tested whether the solvent, Dimethyl sulfoxide (DMSO), used for dissolving small molecules, is compatible with the binding assay. 0–2% of DMSO had no significant effect on the polarization values of the RBD1-numb^FITC^ complex after 2 h incubation (Figure 1D). Upon completion of assay optimization, we carried out a small library screening of small molecules.

### 2.2. Screening of Chemical Libraries

The screening was carried out in a 96-well format. We screened a total of 1920 small molecules from four different libraries (Figure 2A). Z’ factors assess the assay quality and measure the statistical effects; the assay window (ΔmP) calculates differences in polarization value between positive and negative controls. Using the negative (DMSO) and positive (Gn) controls, Z’ factors [35] and assay window were calculated and are shown in Figure 2B,C. We obtained an average Z’ of 0.79 ± 0.05 across all the plates and ΔmP of 74.8 ± 3.7, indicating the robustness of the assay [35]. Using median ± 3SD as the cut off, we obtained 32 hits and a hit rate of 2.03% (Figure 2D). One group of five compounds are sclerotiorin analogues with differences at C-5 and C-7 substituents (Figure 3). Examining the screened library, we identified six more compounds with similar scaffolds (Appendix A), which showed lower inhibition of RBD1-*Numb^FITC^* complex or were completely negative. These six compounds were included in the later validation assays together with the five hit compounds to examine the structure–activity relationship (SAR).

### 2.3. Hits Validation in an FP Dose-Response Test

To further examine the initial hits, we tested compounds with similar scaffolds shown in Figure 3 and Appendix A in an FP dose-response assay. As shown in Figure 4, all compounds except Aza-15 and Aza-17 showed a dose-response effect in disrupting the RBD1-*Numb^FITC^* complex; Aza-9 showed the highest affinity towards GB1-RBD1. We thus focused our further validation studies on Aza-9.

### 2.4. SPR Validation of Aza-9

An SPR assay with the protein GB1-RBD1 and not the RNA allowed us to evaluate the binding of the compound to the protein directly. As shown in Figure 5A, the SPR assay showed a dose-dependent binding of Aza-9 to GB1-RBD1.

### 2.5. RNA Pull-Down of Aza-9

To test whether Aza-9 disrupts the binding of MSI1 to *Numb* RNA in cells, we carried out an RNA pull-down assay using HCT-116 β/W cell lysate. Before we tested our compound in the assay, we tested whether biotinylated *Numb* RNA (*Numb^biotin^*) can pull down MSI1 protein from the cell lysate. We showed that *Numb^biotin^* pulled down MSI1 but *control^biotin^* could not (lane 3 versus lane 4, Figure 5B and Appendix A top panel). Mutations in *Numb* (*Numb-mut^biotin^*) attenuated its binding to MSI1 (lane 2 versus lane 4, Figure 5B and Appendix A top panel). As a positive control, we added non-labelled *Numb* RNA to the system, *Numb* RNA completely abolished the binding between *Numb^biotin^* and MSI1 (lane 6 versus lane 4, Figure 5B and Appendix A top panel). A negative control DMSO solvent for our compounds did not affect the binding (lane 5 versus lane 4, Figure 5B and Appendix A top panel). Next, we tested whether Aza-9 can attenuate MSI1 RNA binding. We showed that 20 μM Aza-9 can attenuate, at least in part, the binding between MSI1 and *Numb* RNA (Figure 5B and Appendix A, lower panel). Our data suggest that Aza-9 binds to MSI1 directly and decreases its RNA binding ability.

### 2.6. NMR Studies of Aza-9

To investigate further the binding of Aza-9 to MSI1-RBD1 and to identify the residues of MSI1-RBD1 that are affected upon binding of Aza-9, we used protein nuclear magnetic resonance (NMR). As described in our previous publication [36], using NMR, we successfully identified the RNA binding residues of MSI1-RBD1—W29, K93, F23, and F65 [37]. Upon titration of Aza-9 with ^15^N-labeled MSI1-RBD1, residues 62, 64, 28, 94, 93, 22, 61, 23, 52, 95, 29, 40, and 68 experienced line broadening, indicating that these residues were affected by the binding of Aza-9 to MSI1-RBD1 (Figure 5C). When mapped onto the structure of MSI1-RBD1, many of the residues previously shown to bind to RNA are directly affected upon titration of Aza-9, along with the residues that are near the RNA-binding regions of MSI1-RBD1 (Figure 5C). Computational docking suggests a binding mode in which Aza-9 interacts with residues K93 and F23 in MSI1-RBD1 (Figure 5D). Binding of Aza-9 near K93 and F23 will compete with the aromatic packing of cognate RNA in the MSI1-RBD1. Our NMR results confirm that Aza-9 binds directly to MSI1-RBD1 and that this binding event occurs in the RNA binding pocket (Figure 5C). In parallel, computational docking provides a model consistent with these NMR observations and allows us to propose a specific binding mode for Aza-9.

### 2.7. Aza-9 Musashi-2 Binding

To test whether Aza-9 can be used as a dual inhibitor to block the RNA-binding ability of both Musashi proteins, we tested Aza-9 for Musashi-2 (MSI2) binding in FP and NMR assays as previously described [28]. We showed that Aza-9 disrupted the binding of MSI2-RBD1 and *Numb^FITC^* RNA in FP (Figure 6A), and it induced reduction of NMR peak intensities when titrated with ^15^N and ^13^C ILV (Ile, Leu and Val) methyl labeled MSI2-RBD1. MSI2-RBD1 residues M23, F64, and K94 showed the most significant reduction in peak intensities (Figure 6B). These residues are in similar positions to F23, F65, and K93 in MSI1-RBD1 [36]. We mapped the peak intensity ratios lower than one standard deviation from the mean (Appendix A) to the structure of MSI2-RBD1. Residues with significant peak intensity reductions clustered around the RNA binding site, which is composed of the central four anti-parallel β sheets and surrounding loops (Figure 6C). The ILV ^13^C methyl groups including I25δ1, V52γ1, and V67 γ2 showed significant reductions in peak intensities, and these residues clustered around the central anti-parallel β sheets (Appendix A). Our NMR results indicate that Aza-9 can directly interact with the RNA binding site, which consists of the central four anti-parallel β sheets and surrounding loops in both MSI1-RBD1 (Figure 5C) and MSI2-RBD1 (Figure 6B) and disrupts MSI1–RNA and MSI2–RNA interactions.

Carrying out the same docking calculation using MSI2-RBD1, we find again that Aza-9 adopts a very similar binding mode as for MSI1-RBD1. Aza-9 is bound near residues K94, M23, and V95 (shown in Figure 6D), consistent with the NMR results. Nearby residues connected through the backbone of the β sheets also include F64 and G26 (Figure 6B), which may respond in the NMR spectra either due to the effect of binding on the nearby sidechains or due to a slight change in the β sheet itself.

### 2.8. Aza-9-Liposome Inhibits Colon Cancer Cell Proliferation, Induces Apoptosis and Autophagy

In an 3-(4,5-dimethylthiazol-2-yl)-2,5-diphenyl tetrazolium bromide (MTT)-based cytotoxicity assay, free Aza-9 had no effect on the cell viability (Figure 7A), potentially due to poor cellular uptake (Figure 7B, upper left panel). We then used PEGylated liposomes to facilitate Aza-9 entry into the cells (Figure 7B, bottom panel compare to upper left panel). Compared to the free Aza-9, which has no effect at 300 µM, Aza-9-liposome (Aza-9-lip) killed colon cancer cell line HCT-116 with an IC50 around 90 µM (Figure 7A). Since liposomes showed toxicity to the cells at 300 µM but not at 100 µM (Figure 7A), we used 100 µM for both Aza-9-lip and liposomes in the following studies. Consistent with MTT, Aza-9-lip, but not liposomes, inhibited cell proliferation in a cell growth assay (Figure 7C). Cell cycle analysis showed that Aza-9-lip treatment also led to the accumulation of cells in G1 phase. (Figure 8A). To determine whether Aza-9-lip induces apoptosis and/or autophagy in the colon cancer cell lines, we first measured caspase-3 and PARP cleavage levels. As shown in Figure 8B and Appendix A, Aza-9-lip induced caspase-3 and PARP cleavage in HCT-116 and DLD-1 cells. Aza-9-lip caused an increase in sub-G1 population in both cells (Figure 8A), indication of apoptosis. Aza-9-lip also induced LC3 conversion and P62 degradation (Figure 8B and Appendix A), indicating autophagy induction and efficient autophagic flux. Taken together, these data indicate Aza-9-liposome inhibits colon cancer cell proliferation and induces apoptosis and autophagy.

### 2.9. Aza-9-Liposome Down-Regulates Notch/Wnt Signaling in Colon Cancer Cell Lines

To investigate whether Aza-9 affects Musashi downstream pathways, we next tested the effects of Aza-9 on Notch/Wnt signaling. As shown in Figure 9A, Aza-9-lip significantly decreased the mRNA level of *AXIN2*, a direct downstream component of Wnt signaling. The levels of *SURVIVIN* mRNA, downstream of the Notch pathway, also decreased in DLD-1 but not HCT-116 cells, whereas SURVIVIN protein was decreased in both cell lines (Figure 9A,B and Appendix A). In addition, the down-regulation of *MSI1* was observed [16,22,36]. To our surprise, Wnt/Notch signaling downstream target gene *CYCLIN D1* (*CCND1*), mRNA and protein expression were increased. Direct targets APC and NUMB protein levels increased in both HCT-116 (Figure 9B and Appendix A) and RKO cells (Appendix A) as a consequence of translation de-repression when MSI function was blocked with treatment. We further tested Aza-9-lip in a functional Wnt reporter assay in HCT-116 cells. Aza-9-lip significantly decreased TOP/FOP reporter signal with (Appendix A) or without (Figure 9C) LiCl stimulation. Taken together, our data suggest that the effects of 100 µM Aza-9-lip on Wnt/Notch pathways are mediated, in part through SURVIVIN inhibition. We did not increase the concentration as higher concentrations of Aza-9-lip would have a more toxic influence from liposomes.

## 3. Discussion

RNA binding proteins are considered “undruggable”, potentially due to lack of a well-defined binding pocket for target RNA [38]. In this study, we used a fluorescence polarization-(FP) based competition assay to identify small molecules that directly inhibit MSI1–RNA interaction. After screening a small library of approximately 2000 compounds, we identified 39 initial hits, five of which are azaphilones. We used FP to test dose response of these five, together with other azaphilones in the library. We further validated one of our top hit, Aza-9, using SPR, NMR and *Numb* RNA-IP, showing that Aza-9 directly binds to MSI1 and inhibits MSI1–RNA interaction. We showed Aza-9 as a dual MSI1/2 inhibitor that inhibited MSI2–RNA interaction as well. In cells, Aza-9-liposome inhibited colon cancer cell growth, induced apoptosis/autophagy and led to G1 accumulation of cells, with a modest down-regulation of Notch/Wnt signaling.

There are reasons why RNA binding proteins are difficult to drug. Unlike many enzymes that have a defined binding pocket employing the “lock and key” scheme, regulatory RNA/DNA binding proteins have shallow binding pockets, and the binding events are not as tight and are less specific due to conformational flexibility. Targeting protein–protein interaction has yielded a list of small molecule compounds that are in clinical trials or in the clinic [39], while inhibitors of protein–RNA/DNA interaction are somewhat new. Our results, presented here as well as our previous publications, suggest that the RNA binding proteins are druggable [14,36,40,41,42,43,44]. In our study, we used an initial FP assay to screen for MSI1 inhibitors, and we validated the hits by three different assays, including SPR, NMR, and RNA-IP. Our assays are suitable for screening small molecule inhibitors of RNA-binding proteins and provide the foundation for large scale screening.

N-terminal RBDs of MSI1 and MSI2 share a high degree of similarity (~87%), thus we tested the binding of Aza-9 towards MSI2 and showed that Aza-9 disrupted the binding of MSI2 to *Numb* RNA in FP and that Aza-9 induced chemical shift perturbations in MSI2. MSI1 and MSI2 share similar roles in stem cells and in cancer initiation and progression, but they also have distinct roles [14,15,45,46,47,48]. Another previously identified Aza-9 target is HuR [44]. HuR is an RNA-binding protein that is overexpressed in a variety of cancers. One defined activity of HuR is to regulate MSI1 [49]. Aza-9 functions to inhibit both HuR and MSI1/2.

Although Aza-9 and the other azaphilones we have tested were produced semi-synthetically from asperbenzaldehyde, our semi-synthesis mimics a natural process. Aza-7 is identical to sclerotiorin, which is produced by a number of fungi [50,51,52], and has been reported to have anticancer activity [52]. Azaphilones are an interesting group of fungal natural products. More than 170 azaphilone compounds are produced by fungi, and they have a number of important biological activities. A substantial number of them have been reported to have anti-cancer activities [53].

With respect to our findings, an obvious question is why would fungi produce natural products that inhibit RNA–protein interactions? Fungi compete with other fungi, protozoans, and bacteria in their natural environments. Many organisms, including bacteria, produce extracellular RNAs that are thought to provide a variety of selective advantages [54]. It is quite conceivable that fungi have developed natural products to combat extracellular RNAs produced by their competitors.

An exciting aspect of the genomics era in natural products research is that we can now over-express genes to produce azaphilones and other valuable natural products. This allows us to produce natural products abundantly and cheaply. Furthermore, by interrupting biosynthetic pathways in organisms such as fungi, we can accumulate large amounts of intermediates such as asperbenzaldehyde and modify them semi-synthetically to increase their potency, efficacy, and medical value.

Investigation of Aza-9-liposome functions showed that Aza-9-liposome inhibited colon cancer cell growth, induced apoptosis/autophagy and led to G1 accumulation of cells, with a modest down-regulation of Notch/Wnt signaling. Our hypothesis is MSI1 and/or MSI2 block the translation of *NUMB*, *APC*, and *P21* mRNA, which leads to the up-regulation of both Notch and Wnt signaling pathways and promotes cell cycle progression. With Aza-9-liposome treatment, Aza-9 binds to MSI1/2, presumably releasing *NUMB*, *APC*, and *P21* mRNA from their translational repression. Increased level of NUMB, APC, and P21 proteins will block Notch/Wnt signaling and cell cycle progression, respectively. In colon cancers, Wnt signaling played a major role in the tumor initiation and progression. Two colon cancer cell lines HCT-116 and DLD-1 used in our study have different genetic profile of *APC* and *CTNNB1* genes that could lead to a different Wnt signaling response. For example, DLD-1 has a truncated APC thus cannot regulate β-catenin encoded by *CTNNB1* gene [55]. However, we observed similar response upon Aza-9-liposome treatment. A study on APC and β-catenin phosphorylation and ubiquitination showed that, although colon cancer cell lines SW480 DLD-1 and HT-29 all have truncated APC, β-catenin ubiquitination and degradation were inhibited in SW480 but not in DLD-1 and HT29 cells [56]. We think although APC is truncated in DLD-1, it can still regulate β-catenin. That’s why we see a similar response upon Aza-9-liposome treatment in both cell lines. However, the same response in two different cell lines might be due to different mechanisms. In HCT-116 cells with a full-length APC, we observed increased APC protein level with treatment, the same increase was observed in RKO cells with the full-length APC. We believe that the increased APC level is responsible for the β-catenin sequestering or degradation, which lead to down-regulation of Wnt signaling. In DLD-1 cells, truncated APC protein levels were decreased slightly upon treatment, 0.97 (50 µM Aza-9-liposome) and 0.89 (100 µM Aza-9-liposome) compared to liposome only (1.00) (data not shown). Other mechanisms in addition to the β-catenin ubiquitination and degradation might account for the decrease in Wnt/Notch signaling we observed in DLD-1 cells. In the future, we will carry our further target validation studies and examination of other MSI targets including but not limited to *SMAD3* and *TGFβR1* [29,57].

In this study, we prepared PEGylated liposomal Aza-9, which facilities cellular uptake of Aza-9. One of our future directions is structure-based optimization of Aza-9 by introducing hydrophilic substituents or shortening the hydrophobic tail, with the aim of improving cell penetration and cytotoxicity to cancer cells with high levels of Musashi family proteins.

## 4. Materials and Methods

### 4.1. Cell Cultures and Reagents

Human colon cancer cell lines HCT-116, DLD-1, and RKO were obtained from American Type Culture Collection (ATCC) and are as described by Lan et al. [36,40]. Cell growth, MTT, cell cycle analysis, western blot analysis, RT-PCR, and quantitative real-time PCR were carried out according to our previous publications [36,58,59,60,61,62]. For western blot analysis with Aza-9-liposome treatment, samples were collected after 24 h; for quantitative real-time PCR, samples were collected after 48 h. The primer sequences, the primary and the secondary antibodies used were from Lan et al. [36]. Fluorescent and live cell imaging was carried out using EVOS FL Auto Cell Imaging System (Invitrogen, Thermo Fisher Scientific, Waltham, MA, USA), and images were cropped and processed using ImageJ (NIH).

### 4.2. Compound Libraries

The chemical libraries used in the initial screening contained three chemical libraries from the National Cancer Institute (NCI) [26] and our own in-house compounds. NCI libraries consist of a (1) natural products set; (2) diversity set II; (3) approved oncology drugs set. Compounds from NCI are stored at −20 °C in 96-well plates at the concentration of 10 mM in DMSO. The in-house compounds are stored as 20 mM DMSO stocks and diluted to 0.5 mM to use in the one-dose initial screening. For the NCI chemicals, 0.1 μL of 10 mM stock was deposited into A2-H11 of each plate in duplicate, 0.1 μL DMSO was placed into A1-H1, and our positive control Gossypolone (10 mM) (0.1 μL) was placed into A12-H12. For our in-house compounds, we first diluted the chemical stocks to 0.5 mM, and then placed 2 μL of each chemical into each individual well. A total of 100 μL RBD1-*Numb^FITC^* complex was added to each well.

### 4.3. Overexpression and Purification of MSI1-RBD1 and MSI2-RBD1 Proteins

pET21a-GB1-RBD1 (MSI1-RBD1) plasmids encoding the *Homo sapiens* RNA binding domain 1 (RBD1, residues 20-107) of MSI1 were constructed with Mus musculus cDNAs under T7 promoter. MSI1-RBD1 proteins were expressed in *Escherichia coli* and purified, as previously described [63]. Protein concentrations were determined using the Bradford assay (Bio-Rad, Hercules, CA, USA). Purification of MSI2-RBD1 protein was carried out according to the previous publication [28,36,40].

### 4.4. FP, SPR, NMR, and Computer Modeling

FP, SPR and NMR assays were carried out according to the previous publication [28,36,40]. For computer modeling of Aza-9 to MSI1/2, the AutoDock4.2.6 program was used for predicting the binding mode. The NMR structure of MSI1-RBD1 in complex with RNA (PDB: 2RS2) and the NMR structure of apo MSI2-RBD1 (PDB: 6C8U) were used as the receptor structure. The MSI1 RBD1–RNA interface and corresponding site in MSI2-RBD1 was used as the starting point for the docking calculations. The initial three-dimensional model of the Aza-9 compound was generated using the BABEL program and was used as starting coordinates to define the ligand structure. For docking, a grid box of size 40 × 44 × 56 A with 0.375 A spacing was centered at residue F23 in MSI1-RBD1 (or F24 in MSI2-RBD1). A total of 200 docking runs were carried out using the Lamarckian genetic algorithm. The docked conformation with the lowest energy was selected as the final conformation.

### 4.5. RNA Pull-Down

RNA pull-downs were carried out according to manufacturer’s protocol with modifications (Roche, Basel, Switzerland). Briefly, HCT-116 β/W cells were plated on 100 mm dish on day 1. The next day, cells were treated with Aza-9 (20 μM) or DMSO for 6 h, and cell lysates were collected using buffer 1 from IP kit (Roche). Protein concentrations were measured for each sample, and streptavidin beads were added for preclearance and incubated at 4 °C for 3 h. After preclearance, for each sample, 500 μg of total protein was transferred to a new tube. For positive control, unlabeled *Numb* (final concentration: 5 μM) was added and incubated in 4 °C. After 0.5 h, heat-shocked biotinylated-*Numb* (final concentration: 500 nM) was added to each sample and incubated at 4 °C for 2 h, after which 50 μL of streptavidin beads were added and incubated for 2–16 h. Beads were washed three times (Roche), and protein sample buffer was added to the beads and boiled for 5 min. Supernatants were taken and loaded in SDS gels for western blot.

### 4.6. Wnt Luciferase Reporter Assay

HCT-116 cells were plated at a density of 4 × 10^4^ cells per well in a 48-well dish the day prior to transfection. Cells were transfected with 0.125 μg of either TOPflash or FOPflash reporter constructs and a pRL-TK Renilla luciferase plasmid to control for transfection efficiency and cell number. Transfections were performed with Lipofectamine 3000 (Invitrogen, Thermo Fisher Scientific, Waltham, MA, USA), according to the manufacturer’s instructions. The next day, cells were stimulated with or without 20 mM LiCl and treated with Aza-9-lip or liposomes control. Cells were harvested and assayed using the Dual-Glo Luciferase Assay (Promega, Fitchburg, WI, USA) 24 h after treatment. All firefly luciferase values were normalized to Renilla control.

## 5. Conclusions

In our effort to screening for Musashi-1 inhibitors, we identified a cluster of hit compounds that are the derivatives of secondary metabolites from *Aspergillus nidulans*. One of the top hits, Aza-9, was further validated by SPR, NMR, and RNA-IP, which demonstrated that Aza-9 binds directly to MSI1–RNA binding pocket. We also showed that Aza-9 binds to Musashi-2 (MSI2) as well. To test whether Aza-9 has anti-cancer potential, we used liposomes to facilitate Aza-9 cellular uptake. Aza-9-liposome inhibits proliferation, induces apoptosis and autophagy, and down-regulates Notch and Wnt signaling in colon cancer cell lines. In conclusion, we identified a series of potential lead compounds for inhibiting MSI1/2 function, while establishing a framework for identifying small molecule inhibitors of RNA-binding proteins using FP-based screening methodology.

## Figures and Tables

**Figure 1 cancers-12-02221-f001:**
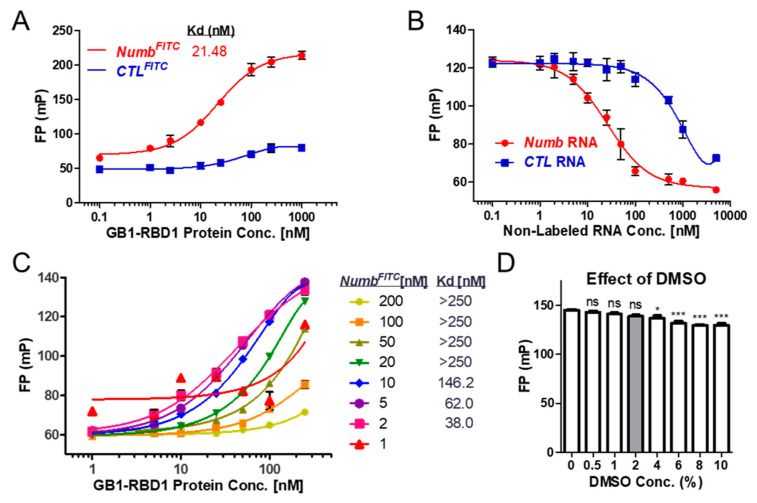
FP assay setup and optimization with MSI1 protein and *Numb* RNA. (**A**) Binding of the RNA binding domain 1 (aa 20–107) of MSI1 (GB1-RBD1) to *Numb* RNA. GB1-RBD1 binds to *Numb^FITC^* RNA (5′-UAGGUAGUAGUUUUA-FITC-3′) but not to control oligo-FITC (*CTL^FITC^* RNA). The concentration of FITC-tagged RNA used in the assay was 2 nM (*n* > 3). (**B**) RNA competition assay. Increasing concentrations of unlabeled *Numb* or *control* RNAs were added to preformed *Numb* RNA-protein complexes. (**C**) Optimization of protein and RNA concentration in FP assay. (**D**) Effect of DMSO on the stability of the FP assay system. ns: not significant; * *p* < 0.05; *** *p* < 0.001 versus no DMSO control (concentration 0).

**Figure 2 cancers-12-02221-f002:**
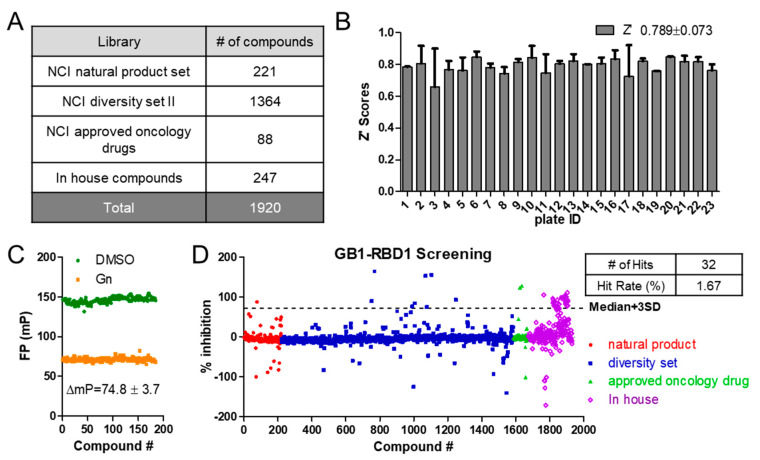
Screening of libraries. (**A**) Library composition. Our MSI1/*Numb* mRNA FP-based screening assay was carried out with 1920 compounds from the NCI (Diversity Set II, natural product set, and approved oncology drugs) and in-house libraries. (**B**) Z’ score across plates. (**C**) Positive and negative controls’ values across plates. (**D**) Scattergram of the screening compounds. Median + 3SD was used as a threshold to pick the hits. Thirty-two hits were identified with a hit rate of 1.67%.

**Figure 3 cancers-12-02221-f003:**
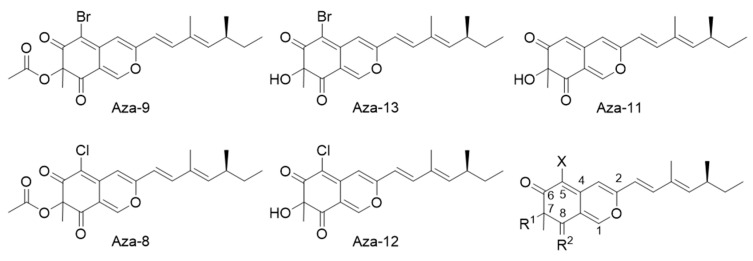
Structures of Azaphilone hits.

**Figure 4 cancers-12-02221-f004:**
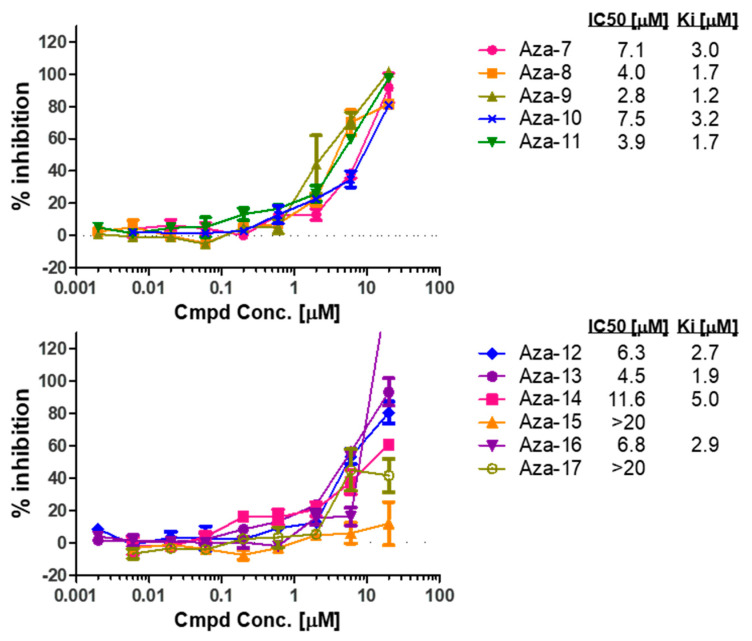
FP dose-response validation compounds binding to RBD1 of MSI1. Ki values were calculated based on the Kd and the dose-response curves.

**Figure 5 cancers-12-02221-f005:**
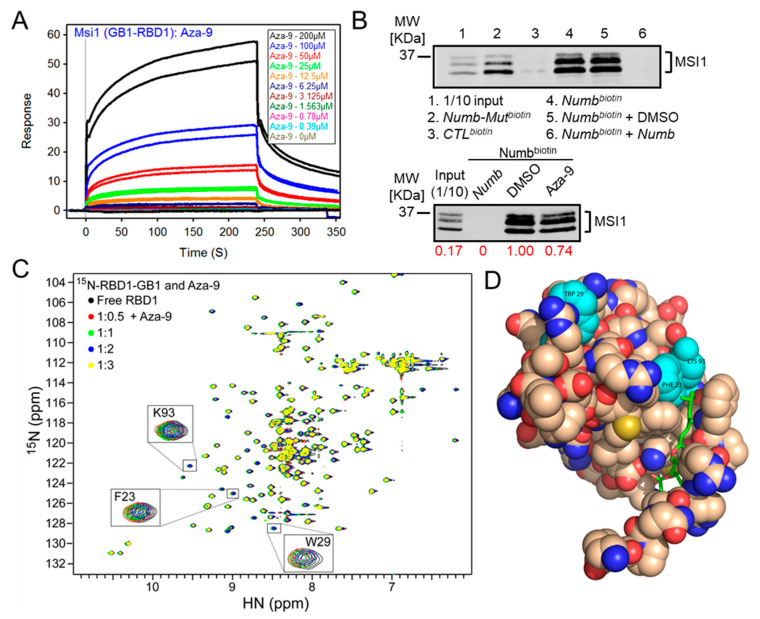
Validation of the top hit Aza-9. (**A**) SPR analyses of Aza-9 binding to immobilized GB1-RBD1. Sensorgram representing direct binding kinetics for Aza-9 are shown in response units (RUs) as a function of time with increasing concentrations (shown as colorful lines). (**B**) *Numb* RNA pulldown. (Top) *Numb^biotin^* (lane 4) can pull down MSI1 but not *CTL^biotin^* (lane 3); *Numb-Mut^biotin^* (lane 2) partially pulled down MSI1. Adding of unlabeled *Numb* RNA (lane 6) abolished the binding completely, while DMSO did not have an effect (lane 5). *n* = 3, one representative western blot is shown. (Bottom) 20 μM Aza-9 attenuated the binding between MSI1 and *Numb^biotin^* RNA. *n* = 2, one representative western blot is shown. (**C**) Overlay of 15N-HSQC (Heteronuclear single quantum coherence spectroscopy) spectra sections of MSI1-RBD1 (black), and MSI1-RBD1 bound to Aza-9 at different ratios. (**D**) Computational docking of Aza-9 bound to RBD1. The Aza-9 is shown in sticks, and the protein is represented as spheres. Three of MSI1-RBD1 RNA binding residues (W29, F23, and K93) that undergo significant peak shifts are highlighted in cyan.

**Figure 6 cancers-12-02221-f006:**
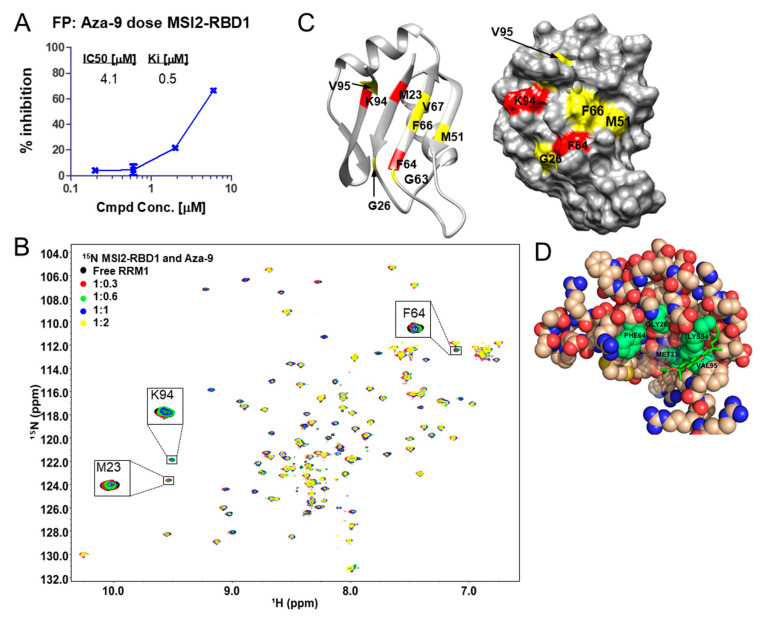
Aza-9 binds to MSI2-RBD1. (**A**) FP dose response of Aza-9 binding to MSI2-RBD1. (**B**) ^1^H-^15^N HSQC spectra of 80 μM ^15^N and ^13^C ILV methyl labeled MSI2-RBD1 with increasing molar ratios of AZA-9. (**C**) Mapping residues with peak intensity ratio lower than one standard deviation from the mean onto the structure of MSI2-RBD1. Residues with peak intensity ratio lower than 0.40 are colored red; residues with peak intensity ratios in the range of 0.40–0.54 are colored yellow. (**D**) Computational docking of Aza-9 bound to MSI2-RBD1. The Aza-9 is shown in sticks, and the protein is represented as spheres. Five of MSI2-RBD1 RNA binding residues (M23, G26, F64, V95, and K94) that undergo significant peak shifts are highlighted in green.

**Figure 7 cancers-12-02221-f007:**
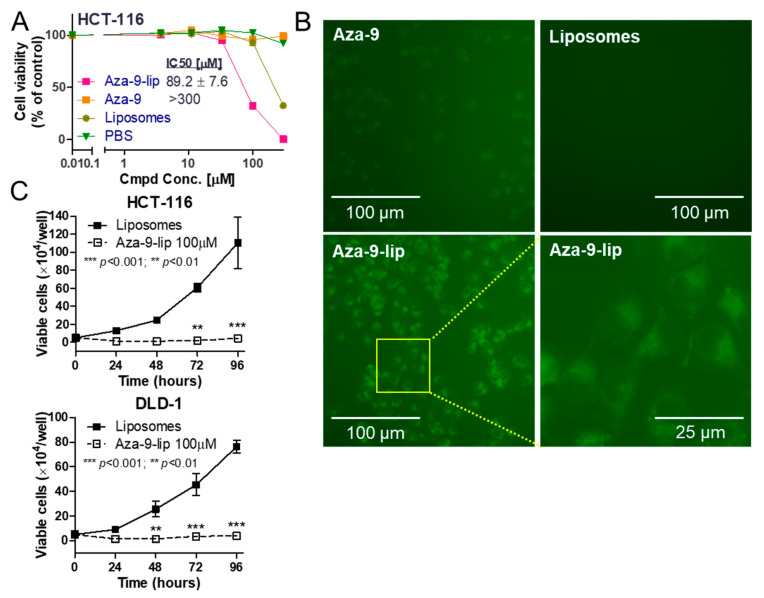
Aza-9-liposome (Aza-9-lip) inhibits colon cancer cell proliferation. (**A**) MTT-based cytotoxicity assay of Aza-9-lip, free Aza-9, liposomes, and PBS in colon cancer cell line HCT-116. Aza-9 had no effect on the cell viability due to poor cellular uptake. (*n* = 3, one representative experiment of three is shown.) (**B**) PEGylated liposomes facilitate Aza-9 entry into the cells. Aza-9 fluoresces in the green channel. Images of Aza-9-lip (100 µM), Aza-9 (100 µM), and liposomes were taken at 2 h after the treatment. (**C**) Aza-9-lip inhibits HCT-116 and DLD-1 cell growth. (*n* = 3, ** *p* < 0.01; *** *p* < 0.001 versus liposomes control. One representative experiment of three is shown.).

**Figure 8 cancers-12-02221-f008:**
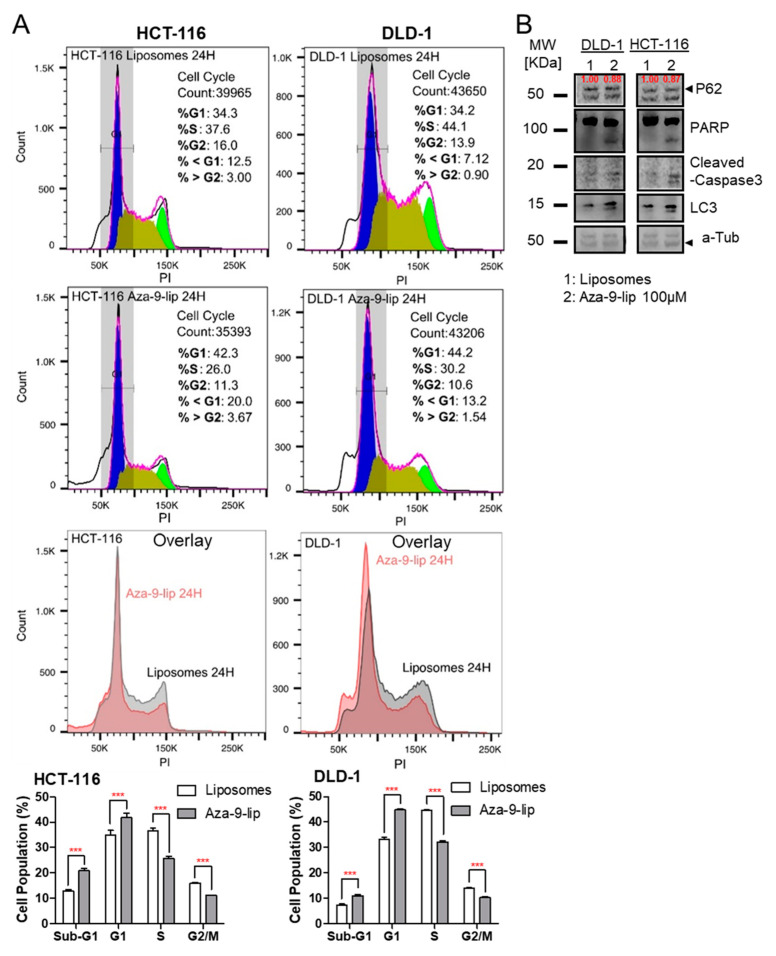
Aza-9-liposome (Aza-9-lip) induces apoptosis (or autophagy) and leads to cell cycle arrest. (**A**) Cell cycle analysis of 100 µM Aza-9-lip or liposomes treated DLD-1 or HCT-116 cells at 24 h. Aza-9-lip treatment led to an increased sub-G1 population, indication of apoptosis. Aza-9-lip treatment also induced a G1 block and led to cell cycle arrest. (Bar graph, *n* = 3, *** *p* < 0.001 versus liposomes control. Spectrums are from one representative treatment.) (**B**) Caspase-3/PARP cleavage, P62 degradation and LC3 conversion were observed in colon cancer cell lines treated with 100 µM Aza-9-lip for 24 h, cell lysate was subject to western blot for PARP cleavage, caspase-3 cleavage, and LC3 I/LC3II expression.

**Figure 9 cancers-12-02221-f009:**
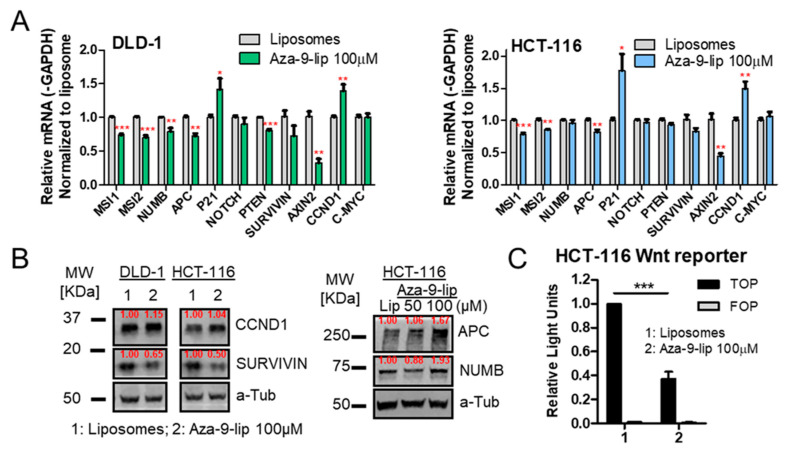
Aza-9-lip down-regulates Notch/Wnt signaling in colon cancer cell lines. (**A**,**B**) Notch/Wnt target genes expression changes upon Aza-9-lip treatment were examined in HCT-116 and DLD-1 cells by quantitative real-time PCR (**A**) and by western blot (**B**,**C**) Aza-9-lip inhibits Wnt/β-catenin reporter. HCT-116 cells were transfected with TOPflash or FOPflash reporter constructs. Cells were treated with Aza-9-lip or liposomes only for 24 h. All figures, *n* = 3; * *p* < 0.05; ** *p* < 0.01; *** *p* < 0.001 versus liposomes control.

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
