# Peer review of "Identification and Validation of an Aspergillus nidulans Secondary Metabolite Derivative as an Inhibitor of the Musashi-RNA Interaction"

_cancers, 2020, doi:10.3390/cancers12082221_

Round 1

Reviewer 1 Report

Lan Lan et al. conducted research to define derivatives of secondary metabolites of Aspergillus nidulans which inhibit the biological activity of Musashi 1 (MSI1) through the suppression of its RNA binding capability. The authors used a comprehensive methodology, including the construction of a fluorescence polarization assay to characterize MSI1 binding to Numb, screening chemical libraries for disruptors. They defined that Aza-9, an Azaphilon compound significantly inhibit the MSI1-Numb mRNA interaction. Further, they validated the dose-response curve, characterized the MSI1-RBD1 RNA-binding residues by NMR, and confirmed the effects at intracellular level using pull-down assay and liposome vehicles. According to their findings, Aza-9 inhibits colon cancer cell proliferation, inducing apoptosis (caspase-3 induction, PARP cleavage) and/or autophagy (LC3 conversion, P62 degradation), and also supressing axin 2 and survivin. They also showed effects at the level of MSI2.

The methodology is correct and complex, the controls are appropriate, the presentation of the results and the conclusions are clear. Their findings are  important, worthwhile to be published.

A few small issues need to adressed before publication:

  1. Abbreviations should be used consistently. In L46, the authors use Musashi-1 (Msi1/MSI1), whereas in L48 they write about msi1 loss-of-function mutants.
  2. There is a confusion in the numbering of the references. Ref no. 45 is cited in the text, but does not appear in the reference list.
  3. Some of the figures, especially Fig. 8 should be improved. The labels are hard to read, the figure is overloaded.

Otherwise, I consider that the manuscript has proper scientific merits to be published in Cancers.

Author Response

1. We apologize for the confusion due to the unclear description of the organisms mentioned. Msi1/MSI1 is for the mammalian systems.  This particular citation in L48 is from a study in Drosophila, we have added the organism to the sentence. 2. We have corrected the reference list.   3. We thank the reviewer for the suggestion; we have reorganized Fig. 8 and enlarged the cell cycle data.

Reviewer 2 Report

Several questions prevent this manuscript from publishing.

(1) Since Aza-9 showed to bind to both MSI1 and MSI2, how could it be concluded that the interference of the MSI1-mRNA interactions is responsible for the reduced tumorigenesis without excluding the possibility caused by the MSI2 binding?

(2) In Fig. 1B, there is a clear competition between RBD1-NumbFITC complex and control RNA even the Ki is lower, indicating that control RNA may bind to GB1-RBD1 which is also shown in their Fig. 1A (even though they claimed "NumbFITC but not CTLFITC bound to GB1-RBD1 as indicated by the increased". This suggest that their experimental design in the first place is questionable.

 (3) in Fig. 2B, it is hard to say that "20 uM Aza-9 can attenuate the binding between MSI1 and Numb" since the intensity of the bands are still very strong. It may require to try a higher concentration of Aza-9 to see if a better and more clear result can be obtained.

(4) NMR followed by computational docking study is actually not a robust evidence to show Aza-9 directly bind to MSI1-RBD1 in the RNA binding pocket. A protein–compound complex crystal structure is a more robust evidence.

 Author Response

1. The reviewer is correct, and we did not intend to give the impression that MSI1-mRNA interactions alone are responsible for the reduced tumorigenesis. We have modified the abstract and the conclusion according to the suggestion, making it clear that binding to MSI2 could be involved. 2. It is not uncommon to see certain limited non-specific binding in a control RNA, however, the specific Numb RNA binding in terms of Kd/Ki suggests strong binding of Numb RNA to the protein, as compared to the control RNA. Using the one-site binding mode calculation with Graphpad Prism software, Kd values in 1B are 25.9 nM for Numb RNA vs 2316 nM for control RNA, a nearly 90-fold difference. 1A and 1B consistently show that the binding of Numb RNA to the protein is highly potent and specific, and the control RNA has limited non-specific binding to the target protein tested. This 90-fold binding selectivity provides a huge detection window sufficient to support a highly specific FP assay used in our study. This is consistent with other reported FP assays widely used in HTS and drug screening. We have used FP systems for drug screening/testing for many target proteins including Bcl-2, XIAP, HuR, Lin28, etc(Nikolovska-Coleska et al. Journal of Medicinal Chemistry, 2004, 47 (10), 2430-2440); US patent 8163805 B2).   3. We have modified our conclusion in section 2.5. We agree with the reviewer that a higher concentration may result in a better inhibition, however the data we present in 5B (not 2B in the comments) show a clear, though partial, competition between Aza-9 and Numb RNA, as supported by the quantification with a 26% reduction. Further optimization of the compound, in vivo PK/PD studies and target validation studies will be carried out. 4. We agree with the reviewer that a co-crystal structure is a more robust evidence, and we will carry out further studies including crystallography and target validation. Unfortunately, it has been challenging to obtain the crystal structure of MSI proteins, which makes it even more difficult to co-crystallize MSI with Aza-9. Further, these studies are beyond the scope of the current manuscript.  Nevertheless, NMR data (Fig. 5C) did show peak perturbations of residues of MSI1 in the RNA-binding region upon titration with Aza-9.  If Aza-9 did not bind to MSI1, the NMR spectra for MSI1 will remain unchanged in the presence of Aza-9.

Reviewer 3 Report

The Authors describe the effect of Aza-9 on RNA binding by MSI proteins. The paper is well prepared, the study hypothesis is clearly presented and all the experiments were properly designed and conducted. Conclusions are supported by the obtained results.

Minor remarks:

The mechanistic evaluation of Aza-9 action in colon cancer cell lines could be improved. I would suggest performing reporter assays in order to assess the effect of Aza-9 on Notch and canonical Wnt signaling in cancer cells. Also, DLD-1 and HCt-116 cells show a different profile of mutations (APC or CTNNB1), which might affect the observed effects on pathway inhibition. The analysis of the molecular mechanism of Aza-9 should be discussed more broadly. It requires elucidation why Aza-9 did not strongly affect the level of target mRNA (Fig. 9A). Was the incubation time 24h? Maybe extending incubation time (e.g. 48h) could show the effects in a more dynamic way. Moreover, what other mRNAs are directly targeted by MSI proteins? 

Please, check and correct references (especially at the end of the list, e.g. 45).

Author Response

We agree with the reviewer that functional reporter assays are important to fully explore the mechanisms of Aza-9 action.  Our current study focuses on the proof-of-principle, and we are planning to carry out more detailed mechanism studies and in vivo tests with more animals to examine this important issue.

For the differences of APC and CTNNB1 in DLD-1 and HCT-116 cell lines that might lead to different wnt signalingresponse, we have not observed any differences upon Aza-9-lipo treatment so far. A study (Yang et al. J Biol Chem 2006, 281, 17751-17757) on APC and β-catenin phosphorylation and ubiquitination showed that, although colon cancer cell lines SW480 DLD-1 and HT-29 all have truncated APC, β-catenin ubiquitination and degradation were inhibited in SW480 but not in DLD-1 and HT29 cells. We think although APC is truncated in DLD-1, it can still regulate beta-catenin. That’s why we see similar response upon treatment in both cell lines. We have added a paragraph in the Discussion section about the molecular mechanism of Aza-9 and discussed about the similar response in DLD-1 and HCT-116.

Aza-9 is not expected to change the mRNA levels of MSI target mRNAs unless such mRNAs are affected by the downstream signaling pathway. The data in 9A was from a 48-hour treatment. We have added the experimental details in the Methods section.

Other mRNA targets include but not limited to CDKN1, SMAD3 and TGFβR1. In the future, we will carry our further target validation studies, examination of other MSI targets.

We have corrected the references.

Round 2

Reviewer 2 Report

Did not see a significant improvement on this revised manuscript as most of the drawbacks are still not fixed. On a side note to the authors, there is a possibility that if the Aza-9 binds to another protein that may cause a cascade effect on MSI1 which will then show a change on its NMR spectra. Therefore, your NMR study results cannot serve as the evidence to prove Aza-9 binds to MSI1 directly.

Author Response

We have addressed the reviewers’ comments and this time we added additional Wnt reporter experiment as suggested. As for the NMR study, there was only one protein present in the system, so the changes on the NMR spectra were from the direct binding event between Aza-9 and MSI1.

Round 3

Reviewer 2 Report

since "there was only one protein present in the system, so the changes on the NMR spectra were from the direct binding event between Aza-9 and MSI1", it just indicates Aza-9 could bind to MSI1 but MSI1 may not be the real binding target of Aza-9 in cells. There are tons of different proteins in cells among which Aza-9 may show affinity towards at various levels. You have to rule out this possibility that what if Aza-9 has a higher affinity towards to protein A, the real binding target while shows lower potency towards MSI1. But since you only put MSI1 as a sole protein with Aza-9 together, even if MSI1 is not the binding target, the NMR spectra will still show some change given Aza-9 having some potency towards it. This is the possibility you have to rule out.